# Food Effect and Formulation: How Soluble Fillers Affect the Disintegration and Dissolution of Tablets in Viscous Simulated Fed State Media

**DOI:** 10.3390/pharmaceutics17050567

**Published:** 2025-04-25

**Authors:** Muhammad Farooq Umer, Valentin Stahl, Jozef Al-Gousous, Thomas Nawroth, Wei-Jhe Sun, Fang Wu, Wenlei Jiang, Zongming Gao, Peter Langguth

**Affiliations:** 1Department of Biopharmaceutics and Pharmaceutical Technology, Johannes Gutenberg University Mainz, 55099 Mainz, Germany; valstahl@uni-mainz.de (V.S.);; 2Department of Pharmaceutical Sciences, University of Michigan, 428 Church Street, Ann Arbor, MI 48109, USA; 3Office of Research and Standards, Office of Generic Drugs, Center for Drug Evaluation and Research (CDER), U.S. Food and Drug Administration (FDA), Silver Spring, MD 20993, USA; 4Office of Testing and Research, Office of Pharmaceutical Quality, Center for Drug Evaluation and Research (CDER), U.S. Food and Drug Administration (FDA), Silver Spring, MD 20993, USA

**Keywords:** food effect, tablet fillers, solubility, media viscosity, fed state, water uptake, porosity, disintegration, dissolution

## Abstract

The food-induced viscosity of the media can alter tablet disintegration and eventually the release of the drug it contains. The extent of this retardation depends on tablet formulation factors, such as the solubility of its excipients. **Objectives**: This research aimed to study the effect of filler solubility on the disintegration and dissolution of tablets under different testing conditions. **Methods**: Tablet formulations containing acetaminophen (as a model compound), mixtures of different ratios of fillers, and other excipients were directly compressed using uniform manufacturing parameters. These formulations were investigated under fasted- and fed-state conditions to determine the influence of viscosity on their disintegration, inspired by the liquid penetration ratio (LPR) theoretical framework. Disintegration and dissolution tests were performed using both compendial and novel testing apparatuses. **Results**: The soluble fillers in the tablets affected their disintegration and dissolution in the simulated fed-state medium, while fasted-state conditions affected the tablets only marginally. The testing devices showed partially contrasting results, which appeared to be due to the hydrodynamics of the testing media used. The novel CNC (computed numerical control) apparatus offered 3D motion and effectively exposed the tablets to the viscous testing media, unlike the compendial paddle apparatus. **Conclusions**: This study explored the impact of filler solubility on the disintegration and dissolution of tablets. As the LPR framework revealed, fillers with a higher solubility have positive effects on the disintegration and dissolution of tablets in viscous conditions. Additionally, the proportion of soluble filler used is also inversely correlated with the disintegration time. Further investigation of the formulation parameters, as well as the testing conditions, would provide additional insights into the effects of food on these tablets.

## 1. Introduction

Immediate-release (IR) oral solid dosage forms constitute the majority of products available on the pharmaceutical market, with a market share of 57.9% in 2017 [1]. This large and important group of pharmaceutical products includes many different formulations. However, all are united by the need for a quick onset of their effects. This is oftentimes governed by the liberation of the active pharmaceutical ingredient (API) from the dosage form via disintegration and dissolution, which are highly correlated but distinct processes [2]. The disintegration of a tablet can be defined as the breakup of compacts into small fragments, which undergo conformational changes upon exposure to a liquid medium [3]. This process represents the initial stage of a cascade of bioavailability, facilitating the breakdown of the tablet to enable rapid drug dissolution [4]. A faster disintegration of the tablet can shorten the release time of the API by rapidly increasing its contact area with the surrounding medium. Conversely, any delay in the disintegration process can have detrimental effects on the overall release of the drug. Some researchers have performed disintegration tests to assess the relevant qualities of tablets, which have provided information on the time it takes for tablets to disintegrate in a given medium [5].

When formulating a tablet product, the goal is to achieve optimal disintegration and dissolution, satisfactory physical and chemical stability, and reproducibility. However, most, if not all, APIs cannot reach these objectives on their own. Consequently, multiple excipients are incorporated into the formulation of tablet products to enable the fulfillment of these objectives [6]. The effective disintegration of a tablet is contingent upon various factors, such as its composition, API characteristics, and the hydrophilic and hydrophobic nature of the incorporated excipients, which collectively govern the disintegration process and the tablet’s subsequent dissolution. Among the excipients are tablet fillers, which comprise a significant proportion of the tablet’s formulation and play a crucial role in facilitating disintegration [7]. The efficacy of the disintegrant used in tablets relies on how much water can reach it, which is influenced by the properties of the filler and disintegrant components [8]. In this study, fillers with different water solubilities were incorporated into tablet formulations to investigate their effects on disintegration and dissolution.

Acetaminophen was selected as the model drug for this study because it has well-defined characteristics and a comprehensive safety profile, and can be formulated into tablets using a variety of methods [9], including direct compression and both dry and wet granulation. As a BCS class I substance it demonstrates high solubility and, as a non-ionic molecule, it is expected to have fewer interactions with the excipients [10]. This indicates that the API’s dissolution is not hindered by its own characteristics, but that the rate-limiting step is its release from the tablet [11]. This process is linked to the properties of the dosage’s form. This study examined the use of a water-soluble compound, acetaminophen, as it could be related to similar compounds from BCS classes I and III.

When testing the properties of tablets in vitro, the choice of media is a crucial consideration, as numerous factors could influence the physico-chemical processes that lead to the tablet breakdown. Various options exist, which range from simply using hydrochloric acid (HCl) solutions to the incorporation of buffer systems and bile salts into complex commercial products. Regardless of the medium selected, the key objective is to simulate the relevant properties of physiological gastrointestinal fluids. In particular, testing tablets in a simulated fed state may require different approaches, with the consideration that the physico-chemical properties (such as pH and lipophilicity) of different media can impact the solubility of specific substances. Previous studies have utilized a blended mixture resembling a typical US American breakfast (FDA-standardized high-fat meal) to mimic the postprandial stomach content of highly viscous chyme [12]. To simulate the desired rheological properties of gastrointestinal fluid in a fed-state condition, a hydroxypropyl methylcellulose-based medium was developed. This simulated fed-state medium has been utilized for various tests, including in studies on compendial disintegration and dissolution devices. Furthermore, a novel apparatus, derived from the Novel Disintegration Apparatus described by Kindgen and Ruiz-Picazo et al., was optimized to accommodate highly viscous media [13,14]. In this investigation, the capabilities of the apparatus were extended to obtain dissolution profiles, providing a comprehensive characterization of the tablets’ performance under these conditions.

The Washburn equation describes the progress of a fluid in a capillary system [15]:(1)$${\left(L\right)}_{cap}=\sqrt{\frac{\gamma r_{pore}t\cos\theta }{2\eta }}$$
where *L* is the penetrating length at time *t*, *r* is the average pore radius, *θ* is the contact angle, and *η* and *γ* are the viscosity and surface tension of the medium, respectively. It is thus evident that the penetration of the medium into the pore network of the tablet is significant in facilitating its wetting and subsequent dissolution. Viscosity is one parameter included in this equation. To anticipate the impact of the soluble components in the tablet formulation under fed-state conditions, a theoretical framework known as the liquid penetration ratio (LPR) was employed. This approach takes into account the properties of both the tablet ingredients and the testing media to predict their impact on the dissolution process.

## 2. Materials and Methods

### 2.1. Materials

The tablet formulations were prepared using acetaminophen (quality according to European Pharmacopeia Ph. Eur. 8.0, Caesar&Loretz GmbH, Hilden, Germany) as a model API. The different tablet fillers were selected based on their degree of solubility in water (Table 1). Fructose (Fagron GmbH & Co. KG, Barsbuettel, Germany), lactose monohydrate (Tablettose 80 MEGGLE GmbH & Co. KG, Wasserburg am Inn, Germany), maltose (Advantose 100, SPI Pharma Septemes-Les Vallons, France), microcrystalline cellulose (MCC, Vivapur^®^ 102, JRS Pharma, Rosenberg, Germany), and dicalcium phosphate dihydrate (DCP, Emcompress^®^, JRS Pharma, Rosenberg, Germany) were used in different combinations. Polyvinyl pyrrolidone K30 (PVP K30, Carl Roth GmbH + Co. KG, Karlsruhe, Germany) was used as the tablet binder, sodium starch glycolate (SSG, Primojel^®^, DFE Pharma, Goch, Germany) as the superdisintegrant, and magnesium stearate (Sigma-Aldrich Chemie GmbH, Steinheim, Germany) as the lubricant. Hydroxypropyl methylcellulose E4M (HPMC, Fagron GmbH & Co. KG, Glinde, Germany) was used to prepare the simulated fed-state media, along with sodium acetate and acetic acid as a buffer. Analytical-grade HCl, sodium chloride (NaCl), and sodium hydroxide (NaOH) were used in this study.

### 2.2. Tablet Formulation

The model drug acetaminophen and the excipients used for each formulation (which are listed in Table 1) were weighed, sieved through a 0.8 mm mesh, and blended in a laboratory mixer (Turbula, Willy A. Bachofen AG, Uster, Switzerland) for 10 min at 34 rpm. Magnesium stearate, which was used as a lubricant, was sieved through a 0.4 mm mesh and added to the mixture, which was then blended for a further 2 min. For each individual formulation, 50 units of round, flat-faced tablets with a constant weight of 650 mg were directly compressed using a 13 mm die and punches in a manual hydraulic press (SPECAC GS15011 Hydraulic press, Specac Ltd., Orpington, UK), which had a compression force of 4 tons and a dwell time of 20 secs. The tablets’ weight after compression was recorded using a weight balance (Mettler PM1200 Giessen, Switzerland). The compressed tablets were given a relaxation time of at least 24 h and kept in the dark and at room temperature in closed plastic containers. In addition to the formulations listed in Table 1, two more formulations were prepared using 25% acetaminophen and valine as “fillers” to further study the effect of solubility on the disintegration time (DT) of these tablets.

### 2.3. Composition of the Testing Media

The tablets were tested in simulated fasted-state and fed-state media [5,21]. The fasted-state medium consisted of a pH 1.2 aqueous solution (prepared by dissolving 2.92 g of NaCl in deionized water and adjusting the pH to 1.2 using 1 N HCl, with a final volume of 1.0 L). In order to simulate the viscosity of food in the fed-state medium, a 1.4% solution of pH 4.5 HPMC E4M was prepared. This entailed dispersing 14.0 g of HPMC E4M in 400 mL of preheated water (80 °C) using a magnetic stirrer and a heat plate. To this solution, 50 mmol of acetate buffer were added, which was prepared by dissolving 1.846 g of sodium acetate and 1.651 g of acetic acid in 100.0 mL of deionized water. The pH of the solution was then adjusted to 4.5 using 1 N HCl or 1 N NaOH, and the final volume was adjusted to 1000.0 mL with deionized water.

### 2.4. Tablet Hardness Measurements

The tablet hardness was determined with a motorized hardness tester (PharmaTest PTB-M Hainburg, Germany), which measured the force needed to disrupt the tablet by longitudinally crushing it between the two platens. The measurements were repeated in triplicate.

### 2.5. Disintegration Test

#### 2.5.1. Compendial Disintegration Test Under Simulated Fasted-State Conditions

All the tablet formulations underwent a disintegration test in a compendial disintegration tester (SOTAX DT2, Sotax AG, Basel, Switzerland), using a six-tube basket without disks and in 800 mL of testing media at 37 °C, in compliance with the USP. Two different testing media were employed: a simulated fasted-state medium and a simulated fed-state medium. The DT for each tablet was recorded visually at a point where no residue was left on the screen. The DTs were reported as means ± standard deviations (n = 6).

#### 2.5.2. CNC Disintegration Test Under Simulated Fed-State Conditions

A secondary disintegration test was performed using a modified novel disintegration tester, which was developed and has previously been used at the Johannes Gutenberg University Mainz [13,14]. The CNC device agitated the samples in a circular, three-dimensional pattern at a velocity of 80 mm/s. Three tablets were tested in parallel. The testing volume was 1500 mL and the temperature was kept within the range of 37 ± 1 °C. The determination of disintegration was performed visually; disintegration was considered complete when no solid residue was left inside the mesh compartment. The DTs were recorded as means ± standard deviations (n = 3).

Selected formulations containing 15% of the soluble fillers were tested using this novel CNC apparatus (F1b, F2b, F3b).

### 2.6. Dissolution Tests

#### 2.6.1. Dissolution Testing in a Compendial Device Under Simulated Fasted- and Fed-State Conditions

A USP apparatus II was used to study the dissolution of all the tablet formulations containing soluble fillers (F1b, F2b, F3b). A total of 500 mL of the fasted- or fed-state media were used in separate vessels at a temperature of 37 ± 0.5 °C. The fasted-state medium was degassed. The rotation speed for both media was set to 75 rpm. Then, 5 mL samples were taken manually at specific time points: 5, 10, 20, 30, 45, and 60 min for both conditions, with additional samples taken at 90 and 120 min for the fed-state condition. The samples were filtered using syringe filters (PES, 5 µm; Macherey-Nagel GmBH & Co. KG, Düren, Germany). The media lost to sampling were not replaced but taken into account during the evaluation of the results. The sink conditions were maintained. The samples were analyzed using UV spectrophotometry, and the results were reported as a percentage release of the total nominal acetaminophen dose (325 mg) and as a mean ± standard deviation (n = 6).

#### 2.6.2. Dissolution Testing in the CNC Device Under Simulated Fed-State Conditions

Dissolution testing was performed analogously to the method used for the disintegration test in the novel modified CNC apparatus. Tablet formulations containing soluble fillers (F1b, F2b, F3b) were investigated. Three tablets were used in each pooled dissolution setup. This experiment was performed in triplicate.

The DT was obtained during the dissolution test. Additionally, 5 mL samples were taken manually at specific time points: 5, 10, 20, 30, 45, 60, 75, 90, 105, and 120 min.

The samples were filtered using syringe filters (PES, 5 µm; Macherey-Nagel GmBH&Co.KG, Germany). The media lost to sampling were not replaced but factored in during the evaluation of the results. The sink conditions were maintained. The samples were analyzed using UV spectrophotometry. The results were reported as a percentage release of the total nominal dose (3 × 325 mg) and as a mean ± standard deviation (n = 3).

#### 2.6.3. Analytics

The amount of acetaminophen in the samples from the dissolution experiment was determined using UV spectrophotometry. The samples were properly diluted for analysis with deionized water and homogenized using a vortex shaker. The solution was left to settle its foam for 20 min and then measured (UV-6300PC) using UV-compatible single-use plastic cuvettes (Carl Roth GmbH + Co. KG, Germany) at a wavelength of 242 nm.

This analytical method was carried out according to the ICH Guideline “ICH Q2(R2) Validation of analytical procedures”. Table 2 provides the experimental parameters used. The standard solution was spiked with HPMC to account for possible matrix effects.

### 2.7. Porosimetry

The porosity and bulk density of the F1b, F2b, and F3b samples were investigated using Mercury Intrusion Porosimetry, using a Pascal 140/440 mercury porosimeter system (Microtrac Retsch GmbH, Germany). The mercury had a purity of >99.9995% and was obtained from GMR Gesellschaft für Metallrecycling mbH, Germany. CD3-type dilatometers were used, which had a filling volume of 450 mm^3^. The measuring sequence was 0–400–0 MPa, with increase and decrease speeds of 6–19 MPa/min. The test was performed with an entire tablet as the sample. The results were corrected for compressibility and evaluated using the software SOLID v.1.6.6., which used cylindrical and plate models.

### 2.8. Working Principle—Liquid Penetration Ratio (LPR)

The impact of the fillers, which is due to their water solubility, was evaluated under simulated fasted- and fed-state conditions. The LPR hypothesis serves as a metric that elucidates the ratio between the penetration rates achieved by two mechanisms: capillary penetration and leaching front. It postulates that the delayed disintegration observed under fed-state conditions could be attributed to the elevated viscosity of the media, which delays the capillary absorption of the fluid into the tablet. The principal mechanism of water penetration in fasted-state conditions resembled capillary uptake through the pore network, analogous to the permeation of water through blotting paper.(2)$${\left(\frac{dL}{dt}\right)}_{cap}=\sqrt{\frac{\gamma r_{pore}\;\cos\theta }{8\eta t}}$$

Equation (2) is a differentiated form of the Washburn equation (Equation (1)), where η is the viscosity of the media, γ is the media’s surface tension, θ is the contact angle, $r_{pore}$ is the effective pore radius, and L is the penetration length at time *t*.

The spatial distribution of the soluble particles can be thought of as a network of interconnected pores traversing the tablet (as long the level of soluble filler in the tablet exceeds the percolation threshold). The infiltration of the surrounding medium into this network can be mathematically represented by employing Fick’s first law of diffusion, as elucidated below:(3)$$\frac{dM_{sol}}{A_{sol}\;dt}=\frac{D_{sol}\;S_{sol}}{L_{diss}}$$
where *M_sol_* is the mass of the solute being diffused, *A_sol_* is its surface area, *D_sol_* is the diffusion coefficient, *S_sol_* is the solute’s solubility, and *L_diss_* is the dissolution penetration length at time *t*.

Dividing Equation (3) by the density leads to the following:(4)$$\frac{dV_{sol}}{A_{sol}dt}=\frac{D_{sol}\;S_{sol}}{\rho_{sol}\;L_{diss}}={\left(\frac{dL}{dt}\right)}_{diss}$$
where *V_sol_* represents the volume of the solute and *ρ_sol_* its density.

Separating the variables of Equation (4), integrating, and applying the boundary conditions at times ‘zero’ and ‘*t*’ gives:(5)$$\int_{0}^{t}dL_{diss}=\int_{0}^{t}\frac{D_{sol}\;S_{sol}}{\rho_{sol}}dt=\frac{L_{diss}^{2}}{2}=\frac{D_{sol}\;S_{sol}}{\rho_{sol}}t$$

The rearrangement of Equation (5) leads to:(6)$$L_{diss}=\sqrt{\frac{2D_{sol}\;S_{sol}\;t}{\rho_{sol}}}$$

Now, differentiating Equation (6) gives:(7)$${\left(\frac{dL}{dt}\right)}_{diss}=\sqrt{\frac{D_{sol}\;S_{sol}}{2\rho_{sol}\;t}}$$

The water-filled cavities formed by the leaching of the soluble filler correspond to the filler particles’ sizes, which are typically much larger than the sizes of the formulation’s pores. Therefore, the leaching front can be viewed as pushing the capillary front forwards, as the leaching front shortens the distance over which the capillary effect is drawing water (as it effectively draws water from the nearest leached cavities rather than from the more distant bulk medium). Accordingly, the *L* in the Washburn equation can be called *L_cap_* (capillary penetration length). Therefore, the sum of the *L_diss_* (dissolution front length) in Equation (7) and *L_cap_* in Equation (2) form the total penetration distance *L_total_*:(8)$${\left(\frac{dL}{dt}\right)}_{total}=\sqrt{\frac{D_{sol}\;S_{sol}}{2\rho_{sol}\;t}}+\sqrt{\frac{\gamma r_{pore}\;\cos\theta }{8\eta t}}$$

Equation (8) shows that the medium penetrates deeper into the tablet due to the leaching of soluble ingredients. Dividing by the differential form of Washburn’s equation allows us to obtain the LPR of a medium advanced by capillary action.(9)$$Liquid\;penetration\;ratio\left(LPR\right)=1+\sqrt{\frac{{4D}_{sol}\;S_{sol}\eta }{\rho_{sol}\;\gamma \;r_{pore}\cos\theta }}$$

In Equation (9), *D*_*s**o**l*_ is the diffusion coefficient of the soluble ingredient, *S*_*s**o**l*_ is its solubility, η is the viscosity of the medium, *ρ*_*s**o**l*_ is true density of the soluble filler, *γ* is the surface tension of the media, *r*_*p**o**r**e*_ is the effective pore radius of the tablet, and *θ* represents the contact angle between the penetrating media and the tablet.

This theoretical framework can predict the general trends in disintegration. Equation (9) reveals that the presence of a highly soluble filler is expected to enhance the medium’s penetration of the tablet and lead to quicker disintegration. This would be particularly true under high-viscosity conditions where capillary uptake is suppressed, which leaves soluble component leaching as the major mechanism by which water uptake occurs (more details in the Discussion section).

## 3. Results

### 3.1. Disintegration Times and Tablet Properties

Table 3 summarizes the DTs recorded with different apparatuses (compendial vs. CNC) and under different conditions (fasted and fed). The weight and hardness of all the tablet formulations were also evaluated. The weight of the tablets was recorded after their compression. The hardness varied among the formulations, as each filler and filler quantity result in the tablets having different properties.

#### 3.1.1. Fasted-State Conditions

The DTs are depicted in Figure 1a. The selection of the fillers used in the tablet formulations was guided by their respective water solubilities. Fructose exhibited the high solubility in water, at 1080 g/L, and was followed by maltose (520 g/L) and lactose monohydrate (230 g/L). Observations under fasted-state conditions (in a low-viscous fluid similar to water) [11] revealed that all the formulations had a quick DT, with a maximum DT of 5 min. The maltose-based formulations (F2a, F2b, and F2c) had a relatively prolonged DT compared with the fructose- and lactose-based formulations. Moreover, an increase in the proportion of maltose in the tablets was found to correlate with an extended DT, as well as with tablet hardness. In contrast, the concentration of the fructose and lactose fillers used in the tablets did not exhibit a significant impact on the DT.

#### 3.1.2. Fed-State Conditions

The DTs recorded under simulated fed-state conditions were found to be significantly influenced by the solubility of each filler. In order to study the effect of the filler solubility on the DT, in addition to the fructose, maltose, and lactose as tablet fillers, valine (65 g/L) and (additional) acetaminophen (24 g/L) were incorporated into additional formulations due to their lower solubility and also tested in this experiment. The results from the compendial apparatus tests depicted in Figure 2 highlight a clear trend connecting decreasing DTs with the increasing solubility of the fillers used in the five tested substances. As valine and acetaminophen are less soluble in water, an increase in the DT can be observed for these additional formulations.

Moreover, the amount of filler in the tablet impacted the DT, which was examined using formulations containing different amounts of fructose, maltose, and lactose. Higher percentages of the fillers resulted in shorter DTs for all three substances (Figure 1b). Fructose (F1), being the most soluble of the fillers, led to the shortest DT at all the three concentration levels. In the case of formulations F2a and F2b, where maltose constituted 25% and 15% of the tablet, respectively, the DT remained the same, at 24 min. However, formulation F2c, which contained only 5% maltose, demonstrated a notably longer DT of 94 min. Overall, the lengths of the F2 DTs were between those of the other two fillers at the three concentration levels tested. Lactose monohydrate (F3), a tablet filler extensively employed in the industry, also demonstrated an ability to reduce the DT as its concentration in the tablet increased. In comparison with the other soluble fillers tested, lactose exhibits a lower solubility, consequently leading to the longest DTs seen among the three fillers at all the concentrations. Formulation F3c, comprising 5% lactose, showed the longest DT of 146 ± 3 min among all the formulations tested in fed-state media. Notably, the DT also displayed an inverse relationship with both the solubility and concentration of the filler used.

For the formulations containing a 15% content of the soluble fillers (F1b, F2b, and F3b), their DTs in simulated fed-state media were also determined using the novel CNC apparatus. Comparing these with the results from the compendial apparatus revealed that all the individual DTs were roughly 2.3–2.8× higher when measured with the CNC device, while maintaining their relative rank order (Figure 3).

### 3.2. Determination of Porosity and Physical Characterization

Table 4 lists the results obtained from the mercury intrusion porosimetry measurements. During the measurements, the sample was compressed, as is indicated by the increase in the density from 0 to 400 MPa, and this was taken into account during the pore size calculations. It can be seen that the bulk densities of the formulations at both 0 and 400 MPa were the lowest for F2b, but, overall, the bulk densities of F1b, F2b, and F3b were similar. F2b had the largest average pore diameter. The total porosity of the samples (given in %) followed the rank order of F1b > F2b > F3b, while the porosities of F2b and F3b were quite similar.

### 3.3. Dissolution Test Resutls

The dissolution tests were conducted on formulations containing 15% concentrations of the soluble fillers. Figure 4a presents their release profiles under both simulated fasted- and fed-state conditions, measured using the USP apparatus II. In general, under fasted-state conditions, all the formulations exhibited faster release rates. F2b, which contained maltose, showed a comparatively slower release than that of F1b and F3b. Conversely, under fed-state conditions, F3b, which contains lactose, showed a rapid initial release that was then followed by a flattened curve, which ultimately resulted in the lowest release rate after 2 h. At this point, the release profiles followed the trend of the fillers’ solubility, with the following rank order: F1b > F2b > F3b. A coning effect was observed with the USP apparatus II, which was exacerbated by the high viscosity of the medium [22].

Figure 4b presents the comparison of the dissolution profiles obtained using the USP apparatus II versus the novel CNC apparatus. The rank order of the formulations’ dissolution rates was not the same for the two devices. In the USP II, the rank orders changed over the course of the experiment. In contrast, in the CNC apparatus, the dissolution profiles maintained the same rank order. Apparently, the fructose (F1b)- and maltose (F2b)-based formulations were more influenced by the changing conditions in the media than the lactose-based formulation (F3b). In the CNC apparatus results, both the fructose (F1b)- and maltose (F2b)-based formulations exhibited a faster drug release, whereas the lactose-based formulation (F3b) demonstrated a slower onset and release, which aligned with these fillers’ solubilities. While this illustrated that there were some differences between the CNC device and USP apparatus II in terms of the dissolution behavior they recorded, the overall trend for the different fillers was similar.

## 4. Discussion

In this study, the focus was on the effect of different soluble fillers on the DT of tablets. From a physico-chemical point of view, the water solubility of all the selected fillers can be considered relatively high. In water, fructose is “very soluble”, whereas maltose and lactose are “freely soluble”. However, there are considerable differences between these fillers (Table 1). The saturation concentration of fructose is about four times higher than that of lactose. These differences are impactful, especially under viscous conditions where capillary penetration and tablet breakdown processes are significantly slower; therefore, the filler’s dissolution is important as a disintegration-promoting mechanism.

The sample tablets were designed such that the total amount of filler was kept constant throughout all the formulations, with the varying soluble filler content being replaced by DCP to make up the remaining amount. By virtue of its lack of swellability and the fact that it is practically insoluble in water, DCP can be considered as a “baseline filler”.

### 4.1. Disintegration and Dissolution Under the Simulated Fasted State

Under fasted conditions, the formulations tested in Figure 1a generally disintegrated rapidly. A similar trend was observed in the corresponding dissolution profiles of the 15% filler formulations (Figure 4a). Under fasted conditions, where low viscosity prevails, capillary penetration through the pore network is the main mechanism behind the uptake of water [5]. This is also known as the wicking effect. The tablet is effectively wetted, and the presence of MCC and superdisintegrant SSG contribute to rapid disintegration, which is primarily driven by the swelling of these excipients [23]. Consequently, no distinct relationship between filler solubility and DT was observed (Figure 1a). Regarding the different filler concentrations tested, in the case of the maltose formulations (F2s), as the filler concentration increased, there was a corresponding prolongation of the DT observed when using the compendial device under the fasted conditions. Similarly, the dissolution profile (Figure 4a) of the maltose formulations demonstrated a slightly slower release despite maltose having an intermediate aqueous solubility compared with the other fillers. However, it still led to the rapid release of the API from the IR tablet. This is possibly due to maltose’s high surface free energy. Therefore, maltose can be considered a saccharide with notable compressibility properties that produces relatively hard tablets (Table 3) with a stronger particle adhesion. This further influences the disintegration behavior of the formulations that contain it [24].

### 4.2. Tablet Disintegration and LPR

There are a number of factors that are responsible for the prolonged DTs observed under fed-state conditions, including tablet hardness, flow velocity, and the viscosity of the media used the represent these conditions [13]. The objective of this research was to study the effect of the solubility of the fillers used in drug formulations. The viscosity of the media plays a crucial role as it influences the shear forces exerted on the tablets, as well as the penetration of liquid into the tablets. Previous studies have reported that a higher viscosity media increased the shear forces on the tablets and thereby facilitated tablet disintegration [25]. On the other hand, viscosity can negatively affect disintegration by limiting the ingress of fluid into the tablet, as it is a parameter in the Washburn equation. Thus, it reduces the availability of water at the target site of the disintegrant and filler [14]. When capillary penetration is slower, fluid uptake via the leaching of soluble fillers becomes a significant pathway for tablet disintegration. This can be explained theoretically as detailed below.

Each parameter in the LPR equation (Equation (9)) has a different impact on the LPR. Media viscosity and surface tension were kept constant for all the samples. In terms of the studied soluble fillers, the densities of fructose (1.69 g/mL), maltose (1.52 g/mL), and lactose (1.54 g/mL) are comparable [26,27]. As soluble fillers are usually composed of molecules that are small and hydrophilic (such as the saccharide units in fructose, maltose, and lactose), their diffusion coefficients are not expected to vary by much. As for the contact angle, switching between different hydrophilic fillers when more hydrophobic components are present, like most lubricants and the majority of APIs, is not expected to profoundly impact wettability. As indicated in Table 4, the tablets’ pore properties were also found to be quite similar (as outlined below). Hence, the greatest variation caused by filler solubility is the extent of the leaching occurring in the samples.

The liquid penetration ratio (Equation (9)) provides a theoretical explanation of the filler solubility effects we have observed, as illustrated in Figure 1. The HPMC medium’s viscosity is more than 100 times higher than that of the HCl media [12], meaning that the low viscosity in the equation’s numerator would result in an LPR close to 1 (i.e., it has little impact on the liquid penetration rate, which stays almost the same as when there is only capillary penetration) even with very soluble fillers, which illustrates that the capillary penetration happens so fast that leaching does not play a significant role. This explains why no correlation between filler solubility and DT was observed in this study. However, the high viscosity observed under fed-state conditions generates LPR values closer to or even higher than 2 for highly soluble materials. An LPR value of 2 means a doubling of the penetration rate, which implies that the penetration rate due to leaching is equal to that of the capillarity. Therefore, an LPR > 2 means that leaching is the dominant penetration mechanism, signifying that it can enhance fluid uptake and partially compensate for the retarding effect of viscosity on capillary movement. Consequently, fillers with solubilities enabling an LPR close to or higher than 2 can accelerate tablet disintegration in a solubility-dependent manner (Figure 5). This explains the strong effect of filler solubility on tablet disintegration seen in the fed state simulated in this study.

In addition to the leaching effect, the higher susceptibility of the leached region of the tablet to erosion due to fluid movement and/or disintegrant action can further shorten the effective distance over which the capillary pressure (as well as leaching) would need to act to draw water toward the tablet core.

Depicted in Figure 2 are the DTs of tablets containing five substances, including soluble “fillers”, which were chosen based on their solubilities. In general, most of these are not commonly used as tablet fillers in the pharmaceutical industry. In terms of tablet disintegration behavior, however, it does not matter if the substance is usually viewed as a sugar, an amino acid, or an API, only its physico-chemical properties determine how it disintegrates. Our selection of fillers gave rise to a clear trend that linked the fillers’ solubility to the tablets’ DT. In order to assess the influence of the measuring apparatus itself, the DTs of the 15% soluble filler formulations (F1b, F2b, and F3b) were obtained using the novel CNC apparatus. Under these altered hydrodynamic conditions, the same trend was observed: a higher solubility was correlated with quicker disintegration. Overall, the DTs determined using the CNC apparatus were longer than those from the compendial apparatus, but the rank order and relative ratios of the different formulations stayed the same (Figure 3). Therefore, the solubility phenomenon observed is likely attributed to the substances’ characteristics rather than the testing method used.

### 4.3. Dissolution Tests Under Fed State

The effect of the fillers’ solubility in the simulated fed state was further characterized using dissolution experiments. The formulations containing 15% soluble filler (F1b, F2b, and F3b) were used for this purpose. In the fed-state medium, their performance also followed a trend that matched the fillers’ solubility. However, it also depended on the apparatus used in the test. The in-house-developed CNC apparatus was designed specifically for testing tablets in highly viscous media. In the USP apparatus II, the tablet was stationary and did not face different hydrodynamic stresses. Hence, the hydrodynamic stress from the media stirred by the rotating paddle in this apparatus caused a distinct coning effect. In the case of the CNC apparatus, the 3D movement of the basket exposed the tablets to the testing media on all sides, which mimicked in vivo conditions and avoided the coning effect [14,28,29].

Under these optimized conditions, there is a clear distinction. F1b released its API the quickest, closely followed by F2b, while F3b was the slowest. In contrast, the results from the USP apparatus II did not fully express this trend, especially during the beginning of the test and at around the 1-h time point. When the experiment was extended to 2 h, the F3b curve flattened, and a solubility-based trend was observed. The rank order of the dissolution rates of the different fillers was not consistent over the course of the entire experiment when the USP apparatus II was used. Apparently, this apparatus can discriminate between fasted- and fed-state release profiles but struggles in distinguishing between formulations tested in fed-state media alone. However, throughout both the extended compendial test and the optimized CNC apparatus test, the influence of the fillers’ solubility was clearly demonstrated to be reciprocally linked to the dissolution of the tablets. This further confirmed the findings from the disintegration tests and underlined the importance of optimized testing conditions for viscous media.

### 4.4. Physical Characterization

The difference in the DTs between the F1 and F2 formulations is smaller than that expected solely based on the filler solubility. The overall disintegration of these tablets is a sum of different factors, with the filler solubility being a major factor, but not the only one. Other parameters, such as physical properties, can influence the disintegration and dissolution processes. In order to further assess the capabilities of the different fillers, the tablet hardness was tested (Table 3). Moreover, the F1b, F2b, and F3b tablet samples (each containing a 15% concentration of their respective filler) were compared using mercury porosimetry. As shown in Table 4, F3b had a slightly higher initial bulk density, but overall, the bulk densities of the three samples were quite similar. The bulk densities at 400 MPa are hypothetical whether the tablets are used in dissolution tests or in patients, as they will not be subjected to such a high pressure in vivo. However, the differences between the formulations’ densities at these two pressures (0 and 400 MPa) indicated the compressibility of the tablets. F1b showed a slightly higher compressed (400 MPa) bulk density compared with that of F2b and F3b. This slightly lower resistance to physical stress was also reflected in the hardness results (Table 3).

As outlined above, intra-tablet liquid transport is dependent on the tablet’s porosity. The rate-limiting step for the wetting of the pore system, which ultimately leads to the tablet’s dissolution, must be identified. As the average pore size calculation method identified the smallest pores as having the largest impact, because they act as a bottleneck for the progression of water through the capillary network; this parameter should be considered in capillary fluid penetration calculations. The average pore sizes of the three formulations were close to each other, with the largest difference not exceeding 25%, while their solubilities varied more extensively. This reinforces that the solubility of these fillers is the main factor impacting the disintegration of the tablets.

### 4.5. Filler Quantity

Not only does the extent of a filler’s solubility influence the tablet’s disintegration behavior, but so does its quantity. Figure 1b illustrates the distinct correlation between a tablet’s soluble filler content and its disintegration characteristics. A larger amount of filler generally leads to quicker disintegration. Notably, the 5% concentration fructose performed similarly to the less-soluble lactose when the latter was used at a concentration of 15%. Increasing the filler content from 5% to 15% seemed to have a larger impact than increasing it from 15% to 25% in the F1 and F3 formulations. This was less obvious with the F2 (maltose) formulations, where both the 15% and 25% samples disintegrated within 24 min. Analogously to the observations made in the fasted state, this can likely be attributed to the tableting properties (such as the high hardness of the tablets during compression) of maltose (Table 3). This can counteract the swelling effect of the disintegrants, resulting in slower disintegration.

The effects of the quantity of the filler used can be explained by the need for the filler level to exceed the percolation threshold for leaching to be able to effectively shorten the distance over which the capillary pressure draws fluid (thus enabling behavior that aligns with the previously described LPR theoretical framework). Beyond this threshold, additional positive effects may come from the greater susceptibility of the leached region to erode or break up due to the action of the agitated fluid and/or the disintegrant.

As outlined above, the tablet filler’s solubility and concentration in the tablet play a significant role in the uptake of water and transport of liquid inside the tablet. The leaching of the soluble fillers creates additional apparent pores in the tablets [30]. Consequently, the highly soluble fructose in F1 formulations extends the pore network more quickly compared with the less-soluble maltose and lactose. However, other characteristics, such as the physical properties achieved by specific manufacturing process parameters, also influence the tablet’s disintegration and dissolution. These will be elaborated on in future studies.

### 4.6. Practical Implications

Acetaminophen, a BCS class I compound, was the compound of choice for our study due to its well-defined characteristics in tablet dosage form, and a direct transfer of these results to other highly soluble drugs (BCS class I and III) should be possible. This implies that these compounds may be administered in many practical scenarios where food intake is optional. As acetaminophen is an analgesic, a quick onset of its effects, regardless of whether the stomach is full, is favorable. This issue is particularly evident with acid-based nonsteroidal anti-inflammatory drugs, such as ibuprofen or diclofenac, which can irritate the stomach and should be taken with food [31]. Some other acute conditions can also be treated with tablets, such as acute glaucoma with the drug acetazolamide. Given the extremely serious nature of this condition, treatment must be initiated immediately and cannot be coordinated with mealtimes. For all these drugs, the delayed effect observed in the fed state, with a slower release from a non-optimized dosage form, would be undesirable.

It is recommended that the antibiotic doxycycline is taken together with food to reduce stomach irritation. There are two forms of doxycycline, doxycycline monohydrate and doxycycline hyclate, which differ greatly in their water solubilities (its monohydrate form is very slightly soluble while hyclate is freely soluble) [32,33]. Tablet products containing either of the two forms are considered generic to each other. Even though the absorption of doxycycline monohydrate is high [34], a disadvantageous tablet formulation could result in its slower release under highly viscous conditions, leading to therapeutic inequivalence.

Antibiotics such as rifampicin, pyrazinamide, or isoniazid should be taken on an empty stomach to ensure their quick and effective absorption. However, in cases of stomach sensitivity or swallowing problems, such as with children, these antibiotics can be taken with food, like porridge or pureed fruit. In this context, differing release patterns in these two different conditions due to food-related compliance are undesirable [35,36,37].

Itraconazole demonstrates a positive food effect and should be taken with a meal to increase its bioavailability [38]. In another scenario, the contents of the meal can influence the absorption of albendazole, as its administration with a high-fat meal results in high bioavailability compared with a low-fat meal [39]. Overall, a predictive tool for optimizing the release of the API from its dosage form in highly viscous environments could be beneficial during the development of drug products.

In the case of poorly soluble drugs, the dissolution of the API can become an additional step that must be focused on after its liberation from its dosage form (depending on the properties of the media, such as its pH). In these cases, aspects of the medium (apart from its viscosity) could be modified by solubility-enhancing additives such as bile salts (which also could impact the wettability of highly dosed, poorly soluble compounds). Investigating the impact of these effects on BCS class II/IV compounds is outside the scope of this work and is a subject for future research.

To summarize, under suboptimal conditions, a non-optimized dosage form could delay the release of a drug in the fed state and cause different fasted- and fed-state disintegrations and dissolutions. In the worst-case scenario, this could lead to the API’s release occurring after its absorption window. The findings of this study serve as a foundation for developing a systematic classification and prediction tool for excipients that considers the role of all the parts of a tablet’s formulation, and not just its API-focused BCS system.

## 5. Conclusions

This study explored the impact of filler solubility on the disintegration and dissolution of tablets. We considered two very different testing scenarios, the fasted and fed states, which mostly differ in terms of media viscosity. In the highly viscous simulated fed state, both the solubility and quantity of the filler used exhibited a notable effect on the DT. The role of both media viscosity and filler solubility are also outlined in the presented theoretical framework (LPR). The more highly water-soluble fillers resulted in faster DTs and API releases in the simulated fed state. Likewise, a larger amount of soluble filler in the formulation accelerated tablet disintegration. In terms of viscous conditions, the hydrodynamic conditions simulated by the novel CNC apparatus are preferable to those of the USP apparatus II. The physical characterization of the formulations using mercury porosimetry showed that they had comparable pore properties, with the small differences in their porosity explaining the slight variations in their performance. Further investigation on the formulation parameters (e.g., different types and ratios of soluble/insoluble fillers, other excipient classes, and manufacturing processes), as well as the adopted testing conditions, would provide additional insights into the effects of food on these tablets.

**Disclaimer:** This study was funded through the FDA, Office of Generic Drugs; Contract 75F40121C00020. The views expressed herein do not reflect the official policies of the FDA or the Department of Health and Human Services, nor does the mention of any trade names imply their endorsement by the US Government.

## Figures and Tables

**Figure 1 pharmaceutics-17-00567-f001:**
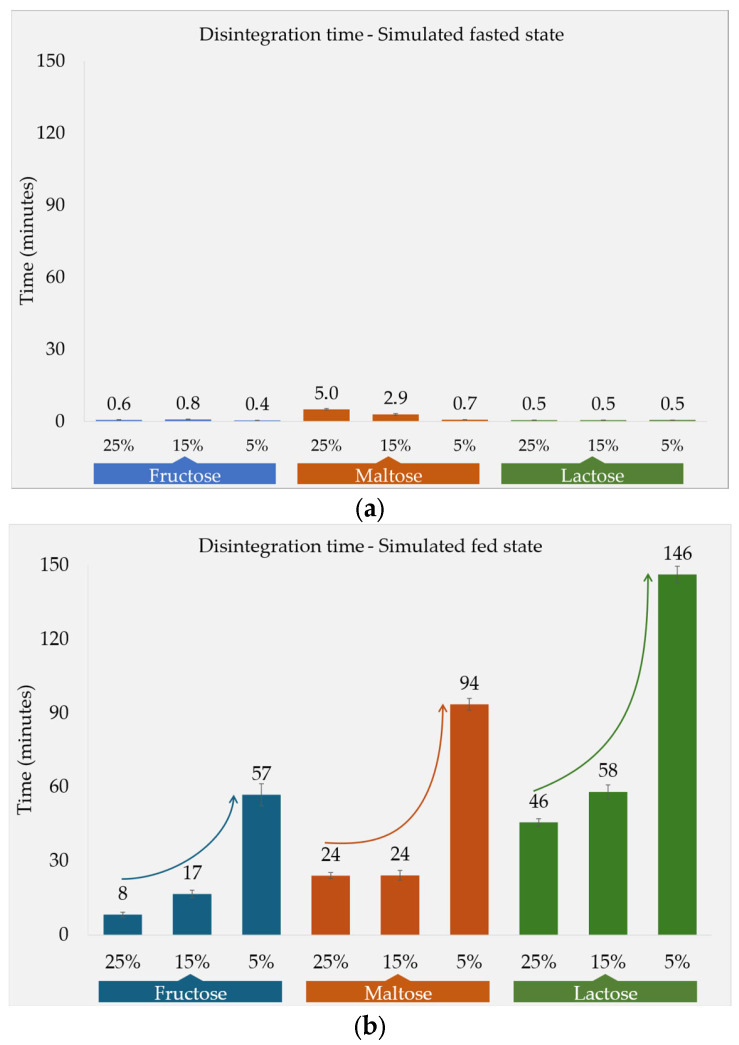
DTs measured with compendial apparatus (**a**) in simulated fasted-state media and (**b**) in simulated fed-state media.

**Figure 2 pharmaceutics-17-00567-f002:**
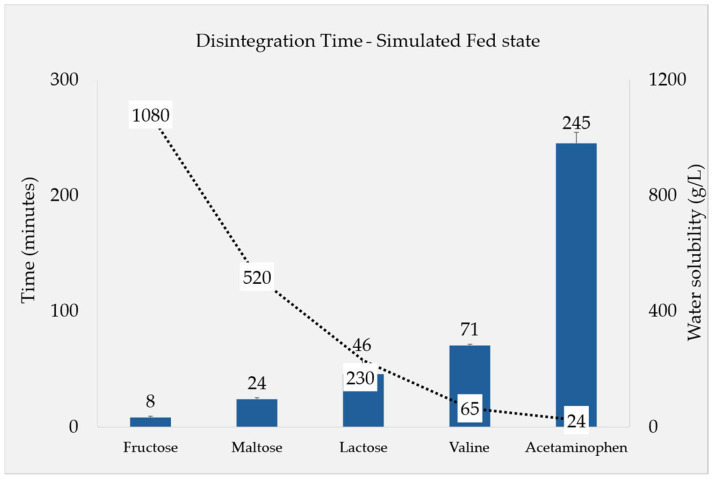
DTs of fillers (columns, on primary y-axis) with different aqueous solubilities (dashed line, on secondary y-axis): all of the formulations contained the same amount of filler (25%), with this extended study using valine and acetaminophen as well (n = 6, Ave ± SD).

**Figure 3 pharmaceutics-17-00567-f003:**
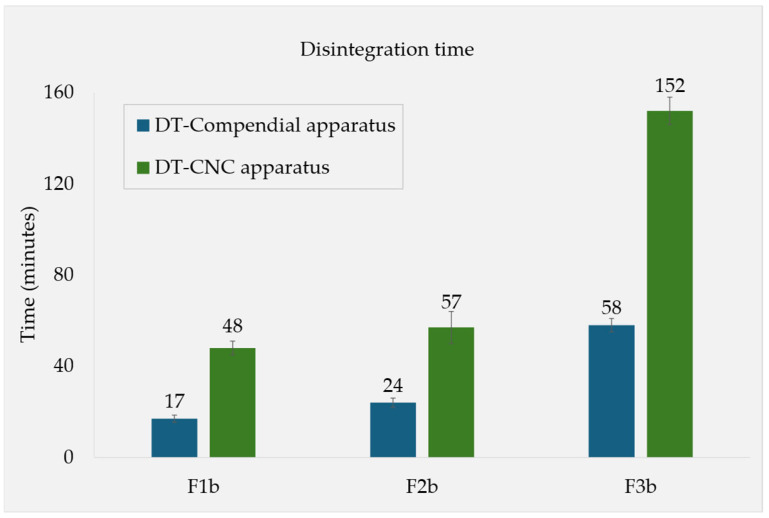
Fed-state DTs measured with different apparatuses. The three formulations F1b, F2b, and F3b contained fructose, maltose, and lactose, respectively, at a 15% concentration.

**Figure 4 pharmaceutics-17-00567-f004:**
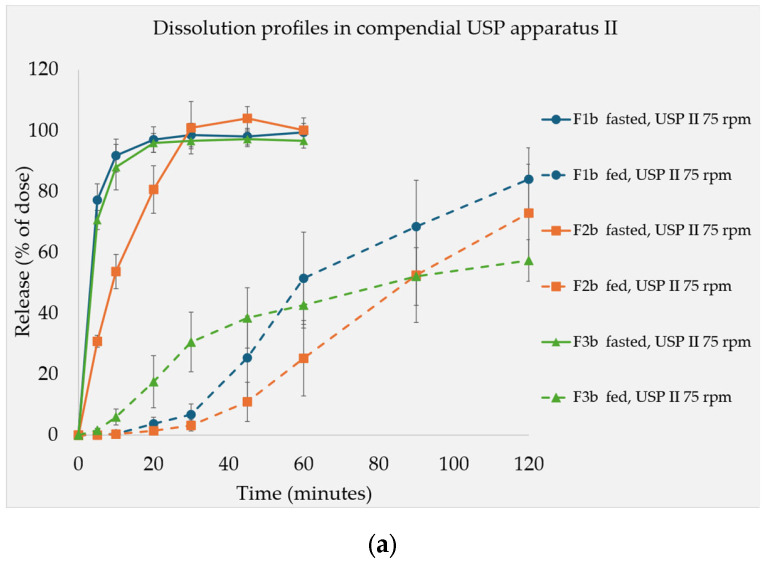
Dissolution profiles of the F1b, F2b, and F3b formulations, which contain 15% concentrations of fructose, maltose, and lactose, respectively, in different media. (**a**) USP apparatus II (n = 6) measurements in simulated fasted- and fed-state media. (**b**) USP apparatus II (n = 6)/CNC apparatus (n = 3) measurements in a simulated fed-state medium.

**Figure 5 pharmaceutics-17-00567-f005:**
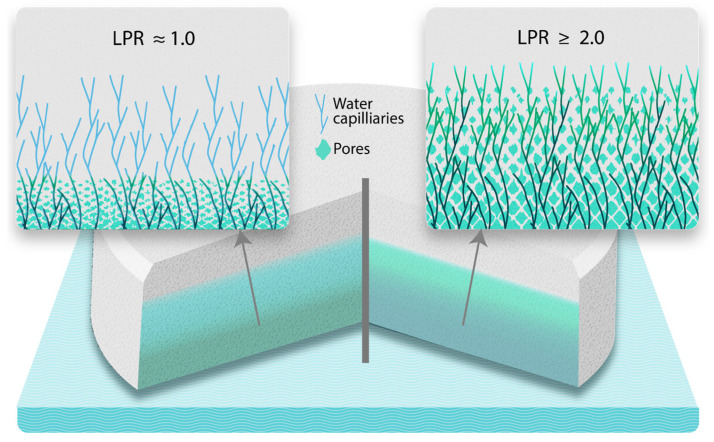
An illustration of unidirectional media transport across a tablet. Two transport mechanisms are represented; namely, capillary penetration alone when the LPR is close to 1, and a leaching front accompanied by a strong “pushing” of the capillary front when the LPR is equal to or higher than 2.

**Table 1 pharmaceutics-17-00567-t001:** Formulation design and filler solubility.

Formulation Code	Percentage Content of Soluble Fillers and DCP Used, 30% of Total Formulation					
	Fructose	Maltose	Lactose	Valine	Acetaminophen	DCP
F1a	25	--	--	--	--	5
F1b	15	--	--	--	--	15
F1c	5	--	--	--	--	25
F2a	--	25	--	--	--	5
F2b	--	15	--	--	--	15
F2c	--	5	--	--	--	25
F3a	--	--	25	--	--	5
F3b	--	--	15	--	--	15
F3c	--	--	5	--	--	25
F4	--	--	--	25	--	5
F5	--	--	--	--	25	5
Water solubility g/L	1080 [16]	520 [17]	230 [18]	65 [19]	24 [20]	Considered insoluble

All the formulations contained acetaminophen (50%), MCC (11%), PVP K30 (5%), SSG (3%), and magnesium stearate (1%).

**Table 2 pharmaceutics-17-00567-t002:** Parameters used for the validation of acetaminophen’s UV quantification.

Parameter	Value
Absorption wavelength	242 nm
Linear range	1–14 µg/mL
Calibration curve	
R2	1.000
Slope	0.0634
Intercept	−0.0001
Limit of detection	0.09 µg/mL
Limit of quantification	0.26 µg/mL
Intraday precision (triplicate)	
4.0 µg/mL	±0.68% relative standard deviation (RSD)
8.0 µg/mL	±0.23% RSD
12.0 µg/mL	±0.53% RSD
Interday precision	
(8.0 µg/mL at three consecutive days)	±0.34% RSD

**Table 3 pharmaceutics-17-00567-t003:** General properties and DTs of the tablet formulations.

Formulation Code	DT—Fasted State, Compendial App., Mean ± SD (n = 6) (Minutes)	DT—Fed State, Compendial App., Mean ± SD (n = 6) (Minutes)	DT—Fed State, CNC App., Mean ± SD (n = 3) (Minutes)	Tablet Weight, Mean ± SD (n = 3)(mg)	Tablet Hardness, Mean ± SD (n = 3)(N)
F1a	0.61 ± 0.11	8 ± 01	*	650 ± 05	64 ± 04
F1b	0.78 ± 0.09	17 ± 02	48 ± 03	651 ± 05	76 ± 02
F1c	0.43 ± 0.03	57 ± 04	*	648 ± 03	92 ± 05
F2a	5.02 ± 0.31	24 ± 01	*	653 ± 03	113 ± 02
F2b	2.87 ± 0.39	24 ± 02	57 ± 07	649 ± 05	87 ± 05
F2c	0.71 ± 0.05	94 ± 02	*	649 ± 02	78 ± 02
F3a	0.52 ± 0.03	46 ± 01	*	653 ± 01	91 ± 06
F3b	0.53 ± 0.03	58 ± 03	152 ± 06	650 ± 02	88 ± 02
F3c	0.53 ± 0.03	146 ± 03	*	650 ± 02	91 ± 04
F4	1.46 ± 0.07	71 ± 01	*	649 ± 01	76 ± 02
F5	0.31 ± 0.03	245 ± 09	*	649 ± 01	41 ± 03

* Values were not obtained using CNC apparatus.

**Table 4 pharmaceutics-17-00567-t004:** Pore characteristics and densities of the F1b, F2b, and F3b tablet samples.

Formulation	F1b	F2b	F3b
Bulk density (g/cm^3^) at 0 MPa	1.2697	1.2630	1.2919
Bulk density (g/cm^3^) at 400 MPa	1.4390	1.3934	1.4069
Average pore diameter (nm)	33.34	41.06	35.32
Porosity (%)	11.67	9.36	8.17

## Data Availability

The original contributions presented in this study are included in the article. Further inquiries can be directed to the corresponding authors.

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
