# Peer review of "Food Effect and Formulation: How Soluble Fillers Affect the Disintegration and Dissolution of Tablets in Viscous Simulated Fed State Media"

_pharmaceutics, 2025, doi:10.3390/pharmaceutics17050567_

Round 1

Reviewer 1 Report

Comments and Suggestions for Authors

Review of the Article: "Food Effect and Formulation: How Soluble Fillers Affect the Disintegration and Dissolution of Tablets in Viscous Simulated Fed State Media" by Muhammad Farooq Umer et al.

This article presents a comprehensive investigation into the effects of soluble fillers on the disintegration and dissolution of tablet formulations under varying viscosity conditions. The authors have conducted a substantial number of experiments using acetaminophen as a model drug, probing how different ratios of fillers and excipients influence tablet performance in simulated fed and fasted states. While the study offers valuable insights, several critical aspects need clarification to establish the relevance of this research within the broader pharmaceutical context.

Main Observation:

Though the authors present extensive experimentation, the rationale behind this work remains somewhat obscure. The paper highlights acetaminophen as a model compound; however, without a clear linkage to the practical implications for other medications, such as antibiotics, the significance of these findings may be diminished. It would be beneficial for the authors to provide a more explicit justification of how the dissolution and physicochemical properties of formulations could affect the bioavailability and overall efficacy of various drugs. As it stands, the connection to real-world applications in pharmaceutical sciences is not adequately addressed. Further experiments comparing different drug classes and their responses to the biopharmaceutical changes induced by food could significantly add value to the work.

Minor Observations:

Several aspects can be improved in terms of presentation and editing. Firstly, the absence of numbering in the subsections makes it challenging to follow the structure of the article clearly. Furthermore, there are stylistic and punctuation errors spread throughout the text, which detracts from the overall professionalism of the manuscript. In Table 3, inaccuracies in the reported uncertainties should be addressed for the data to be reliable. Additionally, the first paragraph of the Discussion section should be deleted.

In summary, while the study provides beneficial data on the characteristics of tablet formulations and their interactions with environmental factors, a stronger emphasis on the real-world applications of these findings is warranted. Additionally, addressing the minor formatting and content-related issues will contribute to an overall improved presentation of this important research.

Author Response

Comment 1: The paper highlights acetaminophen as a model compound; however, without a clear linkage to the practical implications for other medications, such as antibiotics, the significance of these findings may be diminished. It would be beneficial for the authors to provide a more explicit justification of how the dissolution and physicochemical properties of formulations could affect the bioavailability and overall efficacy of various drugs. As it stands, the connection to real-world applications in pharmaceutical sciences is not adequately addressed.

Response 1.

Thank you for your time and effort in reviewing this manuscript and for your valuable comments and suggestions.

A paragraph justifying the use of acetaminophen as a model compound has been added to the introduction. The discussion chapter now includes a subtopic titled ‘Practical Implications’.  This topic now connects our work to real-world applications related to food effects. We also highlighted various drug classes, such as analgesics and antibiotics, along with their food effects relating to drug properties, dissolution medium, and formulation strategies. We have provided some examples that highlight both the negative and positive effects of food and how the drug compound, either by itself or through formulation strategies, particularly the use of excipients, can influence these effects.

Comment 2.

…Further experiments comparing different drug classes and their responses to the biopharmaceutical changes induced by food could significantly add value to the work.

Response 2.

We were expecting this concern. Kindly note that additional drug compounds are currently under investigation within the context of this project. Given that the presentation of those results falls beyond the scope of this manuscript, they are not included here. However, these results are expected to be published shortly, highlighting the application of other drug compounds in relation to the food effect.

Comment 3: …Several aspects can be improved in terms of presentation and editing. Firstly, the absence of numbering in the subsections makes it challenging to follow the structure of the article clearly.

Response 3. That is a valid concern regarding the manuscript's formatting related to numbering. It's important to note that we are using the MDPI template, which, unfortunately, doesn't include numbering for the subsections.

Comment 4: …Furthermore, there are stylistic and punctuation errors spread throughout the text, which detracts from the overall professionalism of the manuscript.

Response 4. To address the issue of textual errors, the entire manuscript has been thoroughly reviewed by a language expert. Along with the resubmission, a proofreading certificate has been included. 

Comment 5: …In Table 3, inaccuracies in the reported uncertainties should be addressed for the data to be reliable.

Response 5: The data in Table 3 has been updated, and the discrepancies have been rectified. The values for the disintegration times in the fasted state (column 2) have been changed from seconds to minutes. The values and standard deviations have been corrected to their respective decimals.

Comment 6: …Additionally, the first paragraph of the Discussion section should be deleted.

Response 6: Thank you for pointing that out. The paragraph was from the MDPI manuscript template, it has been removed.

Comment 7: …In summary, while the study provides beneficial data on the characteristics of tablet formulations and their interactions with environmental factors, a stronger emphasis on the real-world applications of these findings is warranted. Additionally, addressing the minor formatting and content-related issues will contribute to an overall improved presentation of this important research.

Response 7: We appreciate your suggestions on how to relate the current study to real-world applications. For this purpose, we have included the suggested subtopic "practical implications" in the Discussion, which has enhanced the theme of the manuscript. Additionally, a thorough proofreading to address language, grammar, and punctuation has significantly refined the overall quality of the text.

Reviewer 2 Report

Comments and Suggestions for Authors

The manuscript entitled "Food effect and formulation: How soluble fillers affect the disintegration and dissolution of tablets in viscous simulated fed state media" by Muhammad Farooq Umer et al. was well planned and described. My question is - In what medium were the disintegration test and dissolution testing conducted? Why dissolution test ofsome formulations  in figure 4a were provided only for 60 minutes?

Author Response

Comment 1:  The manuscript entitled "Food effect and formulation: How soluble fillers affect the disintegration and dissolution of tablets in viscous simulated fed state media" by Muhammad Farooq Umer et al. was well planned and described.

Response 1: We greatly appreciate the time and effort you dedicated to reviewing this manuscript.

Comment 2: …My question is - In what medium were the disintegration test and dissolution testing conducted?

Response 2: Two types of media were utilised in both disintegration and dissolution tests. To simulate fasted state conditions, a low-viscosity simulated gastric fluid with a pH of 1.2 was employed. For comparison with these fasted state results, a highly viscous 1.4% HPMC solution at pH 4.5 was used to simulate the fed state conditions.

Comment 3: …Why dissolution test of some formulations  in figure 4a were provided only for 60 minutes?

Response 3: These results are from the low-viscosity fasted state medium dissolution experiments, which demonstrate rapid disintegration and dissolution. The dissolution is completed in a short time; therefore, the experiment duration was limited to one hour.

Round 2

Reviewer 1 Report

Comments and Suggestions for Authors

The authors have considered the main points. However, there is one remaining issue, which is the numbering of sections. This is done manually, and we would appreciate it if you could please number the headings.